# Impact of climate change on the distribution of insectivorous bats: Implications for small-scale farming in southern Mexico

Carolina Ureta[1,2,3]*, Mercedes Ramírez-Barrón[1,4], Felipe Ruán-Soto[5], Melanie Kolb[6], Adán L. Martínez-Cruz[7], Giovanna Gasparello[8], Víctor Sánchez-Cordero[4]*

1 Instituto de Ciencias de la Atmósfera y Cambio Climático, Universidad Nacional Autónoma de México, Ciudad de México, México, 2 Investigadora por México-Conahcyt and Member of Laboratorio Nacional de la Biología del Cambio Climático, Ciudad de México, México, 3 Laboratorio Nacional Conahcyt de la Biología del Cambio Climático, Ciudad de México, México, 4 Departamento de Zoología, Instituto de Biología, Universidad Nacional Autónoma de México, Ciudad de México, México, 5 Instituto de Ciencias Biológicas, Universidad de Ciencias y Artes de Chiapas, Tuxtla Gutiérrez, México, 6 Instituto de Geografía, Universidad Nacional Autónoma de México, Ciudad de México, México, 7 Department of Forest Economics, and Centre for Environmental and Resource Economics (CERE), Swedish University of Agricultural Sciences (SLU), Umeå, Sweden, 8 Dirección de Etnología y Antropología Social, Instituto Nacional de Antropología e Historia, Ciudad de México, México

* carolinaus@atmosfera.unam.mx (CU); victor@ib.unam.mx (VSC)

**Data Availability Statement:** All data are contained in the supplementary material.

## Abstract

Bats provide important ecosystem services for agriculture, such as pest control, a function that is particularly relevant for small-scale farmers. However, climate change is causing a decrease in bat populations. To assess the potential impacts of climate change on insectivorous bats and the implications on small-scale farming of indigenous communities in the Chiapas Highlands in southern Mexico we developed a three-step protocol: (1) projecting distribution shifts of insectivorous bats under climate change scenarios using non-dispersal and limited-full dispersal assumptions, (2) using official information to estimate the average economic value of conducting chemical pest control in crops at a state level, (3) surveying small-scale farmers to estimate the local economic value of pest control and determine how bats are perceived by small-scale farmers. Our models project shifts in bat species due to climate change. Given that new suitable climatic areas are also projected, if we assume a limited-full dispersal scenario, bats might not be as at risk by climate change, but shifting their distribution to more suitable habitats will probably affect the dynamic of the ecosystem service they provide. The official estimated value of chemical pest control is $15.15 USD/ha, while the estimated cost resulting from a hybrid survey with small-scale farmers was $47.53 USD/ha. The difference in cost could be related to an overuse of insecticides for pest control or an increase in price due to reduced accessibility. Sixty percent of surveyed farmers perceived a decline in bat populations, 68% were unaware of the benefits that bats provide to their crops, 51% believe that bats are mainly hematophagous, and 10% recognize that people harm or kill bats. A new approach including communicating small-scale farmers in their native languages the benefits that insectivorous bats provide along with a pest management strategy for the efficient use of insecticides needs to be implemented.

**Funding:** Graduate scholarship from Conahcyt The funders had no role in study design, data collection, or analysis and were not involved in the decision to publish this manuscript.

**Competing interests:** No authos have competing interests.

## Introduction

Bats are known to play an essential role in ecosystem services, including pollination, fruit and seed dispersal, and biological pest control [1–5]. Insectivorous bats consume large quantities of insects and contribute to maintaining agricultural production and food security [1, 3, 5–7]. These bat species can limit insect pest outbreaks, induce insect pest migration [7–10], and reduce the environmental and economic costs associated with applying large amounts of insecticides in crops [11]. The importance of species of insectivorous bats in providing ecosystem services by supporting agricultural production is widely recognized [7, 10]. Therefore, it is crucial to determine the vulnerability of species of insectivorous bats to climate change, to high exposure to pesticides, and to people who can harm them.

The Mexican State of Chiapas located in southern Mexico holds a high species richness of 108 bat species (of 140 species nationwide) [12, 13], of which 67 species are insectivorous [14]. Bats have been demonstrated to be sensitive to climate change due to their dependency on environmental cues [14]. Future projections of distributional shifts of insectivorous bats can provide valuable information on how ecosystem services could change regionally. If distributional shifts are projected under climate change scenarios in different time horizons, then ecosystem services provided by species of insectivorous bats might shift correspondingly. On the other hand, small-scale farming (< 5 ha) constitutes an important proportion of the agriculture in Chiapas, where 77% of farmers cultivate maize for subsistence [15].

Chiapas is also known for holding a remarkable biological and sociocultural diversity [16, 17], and it is one of the Mexican States with the lowest GDP [18]. Thus, small-scale farmers with limited economic income are particularly vulnerable to environmental changes caused by climate change that could impact their crop production [19]. However, little is known about how they perceive ecosystem services related to their crops and the corresponding economic impact. Negative perceptions of bats have typically been documented in Asia and South America [20, 21] and in other geographic areas in Mexico [22]. However, in this study, surveyed farmers from several ethnic groups have traditionally interacted with animals [23, 24]. Since pre-Hispanic times, bats have been considered sacred throughout the Mesoamerican region [25]. Ancient Mayan culture viewed them as supernatural beings connected to the underworld and the worship of death; they have also been associated with fertility and the cycle of life [26]. For example, one of the localities included in our study sites is Zinacantán which means "place of bats" in Nahuatl language, and the name of indigenous Tzotziles people who inhabi the study area, comes from "Tsots" meaning "bat" [23, 24]. These traditional communities have foundational registration records that date back to pre-Hispanic times. Their population is composed of indigenous people and mestizos who have lived there all their lives. Their farmlands are located in areas adjacent to where they live, fully integrated into their territory, which positions them to be familiar with the surrounding biodiversity.

We use both quantitative and qualitative methods aiming to provide a more comprehensive understanding of this complex interaction by integrating species ecological information, an economic valuation, and the perception of small-scale farmers of bats as providing an ecosystem service that benefits a better livelihood [27]. We hypothesize that (1) climate change will cause a regional shift in suitable climatic conditions for insectivorous bats that affect their potential ecosystem services; (2) the economic valuation of costs of chemical pesticides based on official data will differ from costs based on the primary information of a hybrid survey conducted on small-sale farmers [see 28]; and (3) small-scale farmers in the Chiapas Highlands have an awareness of the role of insectivorous bats as providing the ecosystem service of consuming insect crop pests. The hybrid survey also provides knowledge on the perception of small-scale farmers on the role of insectivorous bats on their agricultural production.

## Materials and methods

Our methodological approach incorporated a three-stage protocol as follows: (1) using maps generated by a correlative approach to project the distribution shifts of species of insectivorous bats under current and future climate change scenarios in Chiapas, (2) using official information on the use of chemical pesticides on maize fields to conduct an economic valuation through an avoided cost approach, (3) conducting on-site hybrid surveys with small-scale farmers of rainfed fields in Chiapas, which we did for two reasons. First, to conduct a second and more detailed economic valuation of the ecosystem service provided by insectivorous bats and second to gain a better understanding of the perception by small-scale farmers of bats to control insect pests on their crops.

### Ecological niche models

Based on a previous publication, we used the ecological niche models of 67 species of insectivorous bats projected as having their potential distribution restricted to Chiapas [14]. The ecological niche models that were used for this study were originally built based on algorithms that associate environmental factors (climate-related variables) with species georeferenced records, finding an ecological profile—in this case was a climatic profile—that can be identified in the geography [29]. Briefly, following published protocols, only species of bats with a minimum of 25 records (10 km-spaced) were modeled to ensure data for model calibration (70%) and validation (30%) [30]. A set of algorithms, including MaxEnt, GBM, GLM, GAM, CTA, and RF were used, and the validation metrics obtained included Kappa, TSS, and ROC. A total of nineteen bioclimatic variables at a resolution of ~10 $km^2$ (2.5 minutes) were included [31], capturing annual trends and temporal fluctuations of climatic values with biological relevance. For each species, a preliminary variable selection was conducted using a Pearson test ($r < 0.8$). We used species ecological niche models projected to the following scenarios: Baseline (1970–2000), 2030 (2021–2040), 2050 (2041–2060), and 2070 (2081–2100). Furthermore, two distinct general circulation models coming from CMIP 6 [31] were used: CanESM5 (CAN), recognized for its accurate simulation of Mexican climate, and BCC-CSM2.MR (BCC), selected because of its climatic disparity from CAN (being colder and drier). This juxtaposition of contrasting models was aimed at evaluating the climatic variation. Furthermore, each model corresponds to different socioeconomic pathways, with implementations under both SSP2-RCP 4.5 (SSP 245) representing moderate development; and SSP5-RCP 8.5 (SSP585) reflecting a fossil fuel-dependent development trajectory [32, 33].

We used the ecological niche models of the insectivorous bats with records in Chiapas. We then calculated the climatically suitable area gained, lost, and the range change for each species of bats, considering two assumptions: (1) non-dispersal assumption (without considering new climatic suitable areas that might appear in different geographic regions) and (2) limited-full dispersal assumption (considering new climatic suitable areas that might appear in different geographic regions). We used these two assumptions given that there is evidence of species following their climatic niche [34], but we took into account other contributing factors, such as dispersal capabilities and habitat connectivity for species reaching and colonizing new geographic areas [35].

Although not all bat species contribute equally to pest control, we assumed they did due to a lack of specific dietary information in the scientific literature. Unfortunately, the use of mist-netting to identify bat species at the locations under study were logistically complicated due to permission issues with native communities in Chiapas [36].

## Economic valuation

This study implemented the avoided costs method for estimating the economic value of species of insectivorous bats controlling insect pests on maize crops. The avoided costs method infers the ecosystem service value that does not exist in the market through an existing good or service that effectively works as a substitute of the ecosystem services [37]. In our context—for which there are market data of potential substitutes of the pest control service provided by bats —the avoided cost method has a few advantages over other methods of economic evaluation, such as the discrete choice experiment, which has been used in similar contexts in Mexico [38]. For example, the market costs are easily available and help comparability. The avoided costs method is easier to understand and apply, providing more tangible guidance for decision making. When not using the value from the market, accuracy decreases [e.g., 39]. This lack of accuracy tends to increase subjectivity and bias, which makes comparability and decision making more difficult. In this study, given that information is expected to be helpful for decision makers, we decided to use a direct and easy approach [38].

For our economic valuation, we assumed that all species of insectivorous bats equally contributed to insect pests of corn crops, and that they were the only group of animals involved in this ecosystem service. The cost of a single ha of chemical pest control is valued at $15.15 USD (FIRA 2016)—numbers in USD are calculated based on the average exchange rate for the first eleven months of 2023, provided by the Mexican Central Bank (17.81 MXN/USD). To obtain the pest control value of Chiapas using official data we used the following equation:

$$Insect\ pest\ control\ by\ insectivorous\ bats = CHPCH \times SSM \times CHPCUF,$$

where, CHPCH is the cost of chemical pest control per ha in Chiapas ($15.15 USD); SSM is the maize cultivated area (678,590.13 ha), and CHPCUF is the chemical pest control use factor (0.5239%).

## Hybrid survey

We used a hybrid survey to estimate the economic value of pest control based on direct primary information from interacting with small-scale farmers. In addition, the hybrid survey helped us to evaluate the perceptions of small-scale farmers' awareness that species of insectivorous bats are active consumers of insect pests affecting their crop productivity. Specifically, the hybrid survey included a set of questions on a specific topic with answering options [40], and also a direct interaction between the interviewer and the participant whereby no answering options were provided [41]. For our hybrid survey selection process, we used official data to identify geographical cells of 10 km$^2$ with the highest area of maize crop and bat species richness. The localities visited were: Chatetic (Chilón Municipality), Navenchauc (Zinacantán Municipality), San Isidro Chijilté (Teopisca Municipality); and the Municipal headlands of Amatenango del Valle, Teopisca, Venustiano Carranza, Palenque, San Juan Chamula and Socoltenango (Fig 1). These villages have indigenous Tzotzil, Tseltal, and Ch'ol populations as well as mestizo populations, who are native inhabitants in our study sites. At each site, permission was directly requested from each person surveyed/interviewed to conduct the hybrid survey. We also used the data obtained from our study in accordance with local customs and traditions. Only in the municipality of Teopisca was it necessary to obtain permission from the mayor. The snowball sampling method was implemented to identify suitable people to survey/interview [42]. This method identifies individuals who meet specific criteria, such as ethnicity and occupation (traditional peasants). We rely on this sampling method to identify additional participants. The snowball sampling is very useful in qualitative studies to identify key informants who have characteristics and knowledge relevant to specific research, especially in

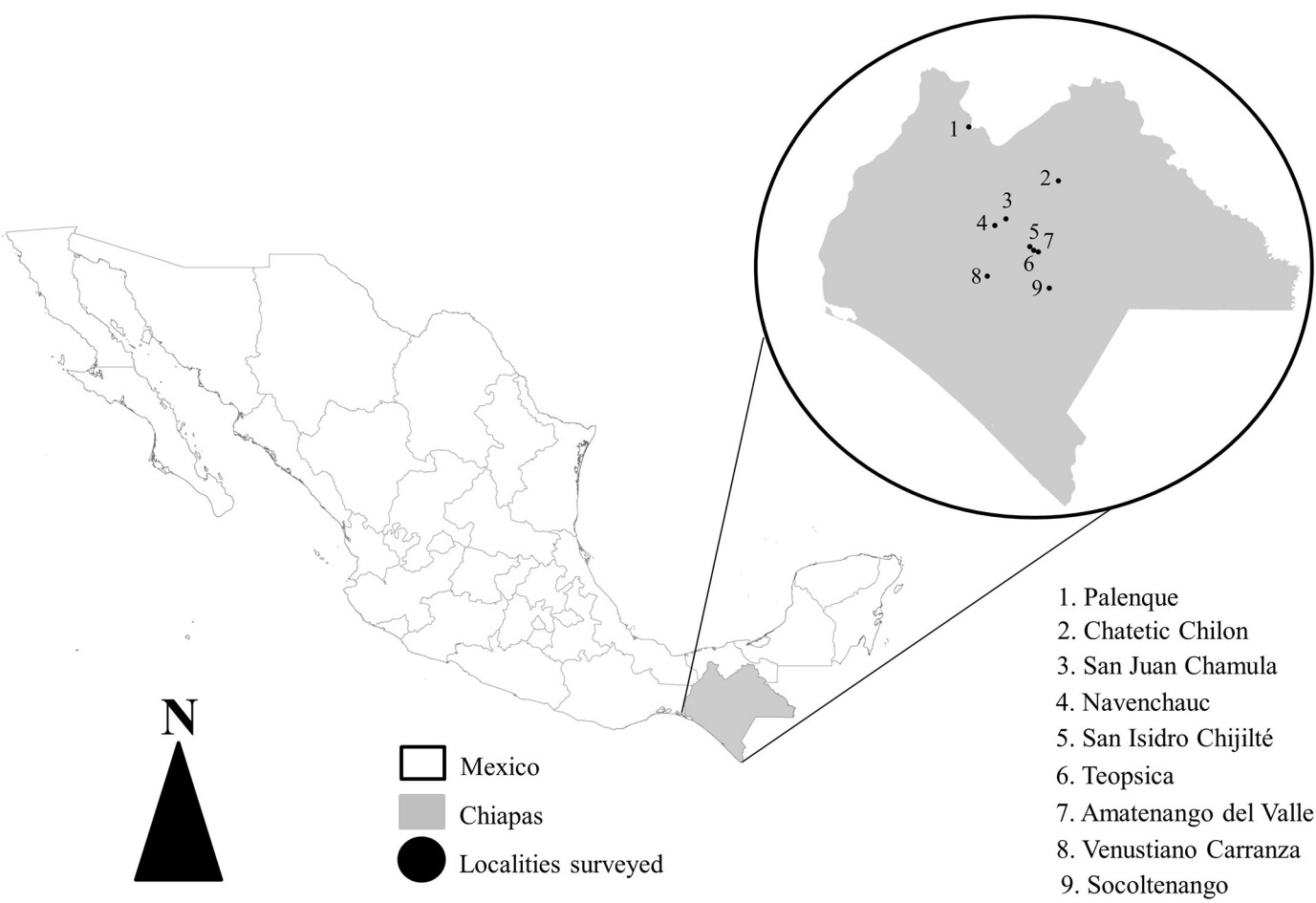

**Fig 1. Study area.** Map depicting the localities where the hybrid survey was conducted. The map of political limits were obtained from https://gadm.org/license.html, where the data are freely available for academic use.

small, bounded, and inhabitants who are sometimes hesitant to collaborate, as is the case of the Chiapas Highlands [36]. A hybrid survey was conducted with the informants allowing a detailed understanding of the perception of small-scale farmers regarding the role of species of insectivorous bats, their methods for pest control and the meaning of growing maize on their field crops. The insights compiled from the answers provided detailed local information about the economic value of the ecosystem service in these communities (S1 File). Furthermore, we applied an avoided-cost method, using the costs involved when farmers apply chemical pest control to their field crops in these localities.

## Results

### Bats distributions models

We used ecological niche models of 67 species of insectivorous bats occurring in Chiapas [see 14] under baseline and climate change scenarios. Under the non-dispersal assumption (without considering new suitable areas that might appear in different geographic regions), the most pronounced suitability declines in Chiapas are represented by SSP585 for time horizon 2070, independently of the general circulation model used. However, stronger climatic

suitability decreases are expected for CAN than for BCC. Also, stronger losses are expected across time horizons. These trends differed under the limited-full dispersal scenario (Table 1, S2 File). For CAN SSP585 in time horizon 2070, 8 out of 67 species showed suitability losses from 90% to 100%, including *Mimon cozumelae*, *Balantiopteryx io*, *Lophostoma brasiliense*, *Thyroptera tricolor*, *Lampronycteris brachyotis*, *Phylloderma stenops*, *Bauerus dubiaquercus*, and *Rhogeessa genowaysi*. The species at the highest climatic risk in most evaluated scenarios were *Balantiopteryx io* and *Mimon cozumelae* (Table 1, S2 File). Overall, the highest suitability losses for most species were observed in the time horizon 2070 and scenario SSP585 characterized by pessimistic fossil fuel emission projections; and greater losses were also expected in the general circulation model CAN representing warmer conditions than BCC.

Under the full-limited dispersal assumption, gains in climatic suitability outweighed losses for all species of insectivorous bats in most scenarios in the general circulation models and time horizons evaluated (Table 1, S2 File). The exceptions were medians under BCC 585 2050 and 2070, and CAN 585 at the time horizon 2070. Only in these exceptions, the climatically suitable area loss was higher than the gains. For CAN 585 for 2070, losses reached nearly 30% higher than climatic gains. The species under higher risk in terms of climatic suitability even when considering suitability gains were *Mimon cozumelae* and *Lampronycteris brachyotis*. Thus, under the two assumptions of dispersal *Mimon cozumelae* was persistently at highest risk. In sum, our results suggest that, when assuming limited-full dispersal ability across Chiapas, most species are likely to find new areas with suitable climatic conditions (Table 1). Overall, a descending trend in climatic suitability over time persists. Despite the presence of the species *Eumops hansae*, *Molossus aztecus*, and *Rhogeessa aeneus* in Chiapas, these three species did not exhibit climatic suitability under the baseline scenario and were consequently excluded from the climate change scenarios projections.

**Table 1. Values of loss and range change in suitable climatic conditions.** The table presents information across three-time horizons (2030, 2050, and 2070) using two general circulation models from CMIP 6: CanESM5 (CAN)—selected for its accurate simulation of the climate in Mexico—and BCC-CSM2.MR (BCC)—chosen for its climatic disparity from CAN, being colder and drier. The table includes all species of insectivorous bats evaluated. Two different socioeconomic pathways were considered: SSP2-RCP 4.5 (245), representing moderate development, and SSP5-RCP 8.5 (585), reflecting a fossil fuel-dependent development trajectory. The analysis was conducted under two different dispersal assumptions: Non-dispersal and Limited-full dispersal. Values under the Non-dispersal assumption indicate the percentages median of pixels lost in the study area (~10 km$^2$ (2.5 minutes)). Values under the Limited-full dispersal assumption represent the range change median, accounting for both areas lost and gained, respectively. Bold numbers correspond to minimum and maximum values.

**Non-dispersal assumption**

|  | 2030 | 2050 | 2070 |
|---|---|---|---|
| 245 BCC | **-0.173** | -0.265 | **-0.25** |
| 245 CAN | **-0.113** | **-0.179** | **-0.25** |
| 585 BCC | -0.143 | **-0.316** | -0.375 |
| 585 CAN | -0.143 | -0.256 | **-0.452** |

**Limited-full dispersal assumption**

|  | 2030 | 2050 | 2070 |
|---|---|---|---|
| 245 BCC | **0.032** | 0.085 | 0.016 |
| 245 CAN | 0.111 | **0.095** | **0.048** |
| 585 BCC | **0.114** | **-0.80** | -0.078 |
| 585 CAN | 0.085 | 0.022 | **-0.271** |

## Hybrid survey

A total of 77% of farmers in Chiapas are small-scale farmers who cultivate maize for subsistence [15]. In March 2020, we conducted 53 hybrid surveys with small-scale farmers from various ethnic groups in Chiapas. Although the sample size needed can be controversial, it has been considered that between 20 and 60 is a good enough number of people to be interviewed in a qualitative study [43], and over 50 participants can be considered an acceptable limit size even when the selection could not be random and independent [44]. Our hybrid survey included 58% Tzeltales, 25% Mestizos, 15% Tzotziles, and 2% Choles. Of those surveyed/interviewed, 79% were male and 21% female (Table 2). Half of the interviewed farmers reported using a similar portion of their maize harvest for personal consumption and for selling: Approximately 36% of respondents stated that their maize harvests are solely for self-consumption, 13% of farmers cultivate maize exclusively for selling, and 51% consume and sell the maize they cultivate (S3 File).

Furthermore, 22% of small-scale farmers mentioned that they have observed bats in their crops and most farmers believe that bats feed from blood or fruit (> 90%). In addition, 10% believed that bats could be eating the maize crop. Only two respondents recognized that bats could be eating insects, while another two respondents recognized that they might be pollinating maize from crops. Most farmers (68%) did not perceive bats as beneficial (Fig 2). The Tseltal farmers (40%) were the least familiar with the role of bats as consumers of insect pests, whereas Mestizo farmers (11%) were the most aware of the benefits of bats eating insect pests (Table 2). Further, when we asked farmers if they had observed more or fewer bats in the last year compared to previous years, 73% indicated that they had observed fewer bats in the area. In contrast, 21% perceived an increase in the bat population, while 15% mentioned that they had not seen any bats, and 2% believed that the number of bats remains the same. About 10% of the farmers recognized that the number of bats could be varying because they are getting harmed or killed. The farmers attribute the fluctuation in the bat population primarily to deforestation, climate change in the region, and human-induced mortality.

When asked about purchasing more pesticides due to an increase in insect pests in corn, 30% of farmers expressed the ability to afford such products for pest mitigation, 13% were uncertain about their ability to purchase, and 57% mentioned they would not be able to afford the costs of pesticides. Those who indicated they could not purchase more pest mitigation pesticides mentioned that they would have to seek for loans, find an alternative to save money, or even consider not planting corn at all (S3 File). Lastly, we inquired about the farmers' responses if they had to stop cultivating maize: 36% of farmers answered that they would try to cultivate a different crop, while 32% of farmers mentioned they would not have access to food. Some farmers considered the option of changing their occupation (15%) or purchasing maize (7%), but others were considering abandoning their land (6%) (S3 File).

**Table 2. Percentage of indigenous small-scale farmers surveyed/interviewed, categorized by language, who are aware (% Yes) and unaware (% No) of the ecosystem service provided by species of insectivorous bats in pest control of corn crops in the sampled localities in Chiapas.**

| ISF | Yes (%) | No (%) | Do not know (%) |
|---|---|---|---|
| Ch'ol | – | – | 2 |
| Mestizo | 11 | 13 | – |
| Tseltal | 6 | 40 | 13 |
| Tzotzil | 2 | 11 | 2 |
| **Total** | **19** | **64** | **17** |

# What do bats eat?

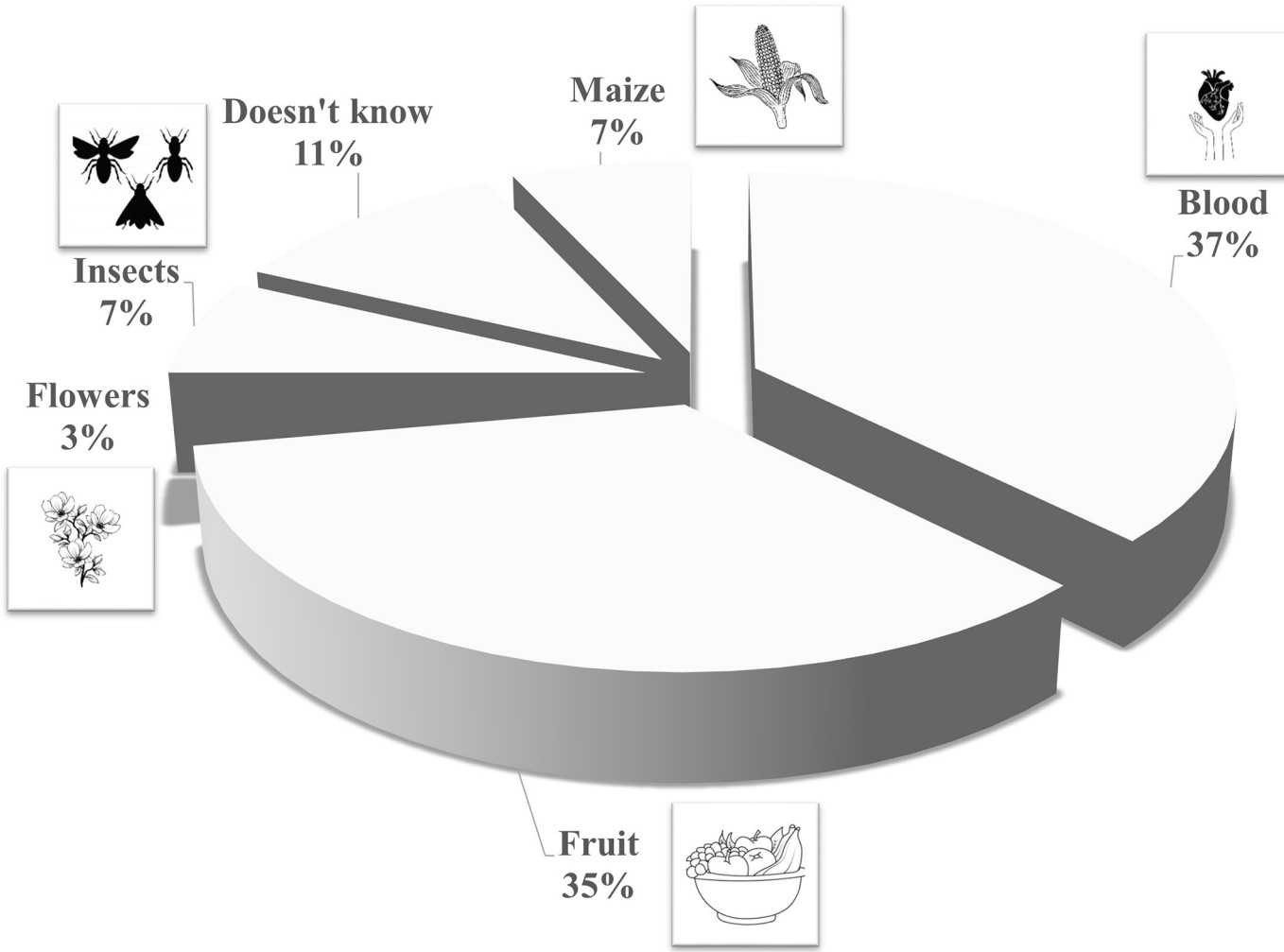

**Fig 2. Responses provided by small-scale farmers regarding what they believe bats consume to understand their awareness of pest control that bats provide.**

## Economic valuation

The area cultivated with maize in Chiapas is 678,590.13 ha, and only 52.39% of the entire land surface (355,513 ha) reports using chemical pest control [45, 46]. The official information estimates that an average of $15.15 USD/ha for pest control each year. Thus, the estimated value for pest control at the state level is $ 5,386,027.54 USD. Based on the hybrid survey, we estimated that small-scale farmers spent about $4,950 USD per year on chemical compounds for pest control to cover 104.1 ha of maize. Thus, small-scale farmers spend about $47.53 USD/ha in one agricultural year. If we extrapolate this value to the entire agricultural surface cultivated with maize in Chiapas, the cost of the chemical pest control is $33,798,265.36 USD. If we assume that only 52.39% of the agricultural surface of Chiapas uses chemical pest control [45], then the total estimated cost is $16,899,132 USD. Our findings also revealed that 81% of small-

scale farmers employ chemical pest controls in maize crops, 15% do not use any form of pest control, and 4% use biological control methods (such as fish waste or chili-infused salt).

## Discussion

Future rising temperatures are likely leading to an increase in the number of insects infesting maize crops, highlighting the importance of insectivorous bats as pest control [47]. Our study found that insectivorous bats in the Chiapas Highlands are likely to shift their distributions to more suitable areas in the coming years. Distribution shifts driven by changes in climatic variables have already been documented for several mammal species [34]. On the other hand, species of insectivorous bats are expected to show strong reductions in their suitable climatic habitats in the Chiapas Highlands under a non-dispersal assumption (Table 1; S2 File). Insectivorous bat distributions under climate scenarios will likely have negative impacts on the ecosystem services that these bats provide as active consumers of insect pests on maize crops in the study site (Tables 1 and 2; S2 File). Interestingly, shifts in species distribution projected under baseline and climate change scenarios coincide with the perception of small-scale farmers of identifying fewer bats visiting their maize crops in recent years (Table 1; S3 File). We believe that this signals an opportunity for continued monitoring of insectivorous bat species in the Chiapas Highlands to determine which species and at which rate their distributions are shifting from the Chiapas Highlands to new climatically suitable habitats. It would also be recommended to include dietary monitoring due to the lack of specific dietary information for bat species in the area. DNA metabarcoding could also be valuable in assessing changes in consumption patterns and their relation to climate change [48]. Additionally, it could clarify other perceptions of farmers, such as bats feeding on maize. The monitoring program should be established along with quantifying crop damage due to insect pests on maize cultivation, and the potential economic and social consequences for small-scale farmers.

Furthermore, from a conservation perspective, there are several challenges associated with the need for insectivorous bats to shift distributions to new habitats of suitable climatic conditions. We observed that the species showing a higher risk under climate change scenarios in Chiapas was *Mimon cozumelae*. This bat is not endemic to Chiapas, although it is considered to be a threatened species under the Mexican legislation and is highly vulnerable to changes in environmental conditions [49]. Another conservation risk relates to an overuse of chemicals for pest control. Considering that the costs associated with using chemicals for pest control were more expensive when values were obtained through the hybrid surveys than with official data suggests that small-scale farmers in Chiapas Highlands are using more than expected. An overuse of chemicals for pest control on crops has already been documented in other small-scale farmers elsewhere nationwide and in other countries [50, 51]. An abuse of chemical pest control can lead to increased health risks for farmers and their families, bats, and wildlife in general [52, 53]. Although many species of bats can move along agricultural landscapes [54], exposure to chemical pests poses a growing threat increasing the vulnerability to health risks [55, 56]. For example, use of pesticides in crops has been linked to declines in bat populations due to poisoning [57].

In addition, small-scale farmers pay more for pesticides probably because they need to be transported to their remote and isolated communities [58]. The cost of chemical pest control for small-scale farmers is more than three times higher than the costs obtained by official data (Tables 1 and 2). Thus, small-scale farmers in the Chiapas Highlands heavily rely on species of insectivorous bats to decrease the insect pest populations that damage their maize crops or pay high costs for chemical insecticides for pest control, which is a burden to their limited income (S3 File). This scenario highlights the complex interplay of climate, biodiversity conservation,

ecosystem services, and socioeconomic factors on crop production, and urges for integrated pest control management in accordance with the conservation of insectivorous bats in this region. This is particularly relevant for small-scale farmers with very limited economic resources.

The cost of chemicals to control insect pests purchased by farmers allowed us to quantify the value of the ecosystem service provided by bat-mediated insect pest control as a partial substitute for chemical control methods. Calculations using available official data yield $15.15 USD/ha as the value of bat-mediated insect pest control in Chiapas. However, calculations based on data reported by farmers participating in our study yield an estimate of $47.53 USD/ha. This difference in estimates between official data and primary data arising from our hybrid survey is consistent with previous evidence documenting that small-scale farmers in Mexico, especially in southern regions like Campeche and Chiapas, lean toward overuse of pesticides and fertilizers, particularly in wheat and maize production [59–61]. In general, however, the value arising from primary data is higher in comparison with other geographic regions. For example, Gándara et al. [62] estimated that the cost of pest control in maize ranges between 8.3 and $18.6 USD/ha in Nuevo León, Northern Mexico, based on pesticide applications. Maine & Boyles [63] assessed bat-mediated pest control at $7.9 USD/ha in the United States. On the other hand, Aguiar et al. [64] found that the ecosystem service value of insect pest suppression by insectivorous bats saves farmers $94 USD/ha, reaching an estimated annual savings of $390.6 million per harvest in Brazil. These studies are in accordance with what can be inferred with official information although they lack the firsthand information that our study obtained directly from small-scale farmer families cultivating maize in Chiapas. The information obtained by our economic valuation provides support for the hypothesis that small-scale farmers spend more money on pest control than the average farmer, whose expenditure calculations come from official data from Chiapas.

We anticipated that small-scale farmers were more sensitive to the importance of species of insectivorous bats given the familiarity they have with the environment. However, half of the people (27 of 53) interviewed from the ethnic groups by our hybrid survey considered bats as harmful (Fig 2, S3 File). For example, they raised concerns given that they believed that most bats were hematophagous. This argument is supported by the fact that small-scale farmers relate the decreasing presence of bats with a decrease in sheep in the study area, perhaps linking bats feeding on cattle (S1 and S3 Files). This misperception represents a challenge to promoting bat conservation. One alternative is to design programs offering farmers compensation —either monetary, non-monetary or both—in exchange for a reduction in use of chemical fertilizers and pesticides to benefit bat populations, along with technical support on how to manage pests more ecologically. For example, small-scale farmers cultivating *Agave* plantations to produce tequila are under a program of economic compensation if they take measures aimed at improving the genetic conservation of blue agave and the protection of the lesser long-nosed bat *Leptonycteris yerbabuenae* [65]. Conservation strategies of bats in our study area can be promoted by incentivizing the remaining 50% of small-scale farmers that perceive bats as not harmful, with science-based information on the importance of species of insectivorous bats as ecosystem providers of crop pest controllers. This effort will likely result in convincing a high proportion of small-scale farmers of the benefits that bats provide humans [1–5].

The lack of awareness of small-scale farmers of the importance of insectivorous bats to provide ecosystem services shows a cultural erosion among the ethnic groups surveyed/interviewed. The various colonial domination processes and the 20th-century indigenist policies discredited the knowledge of indigenous people and prohibited its reproduction. This situation might have broken the chain of intra- and intergenerational transmission of knowledge related to the territory. That is probably why our hybrid survey provided enough information

to assume that most small-scale farmers of the Chiapas Highlands are unaware of the importance of insectivorous bats. We also detected a language barrier between farmers belonging to different ethnic groups, hindering access to information about the importance of insectivorous bats. Only Mestizo farmers readily recognized the benefits provided by insectivorous bats, possibly due to their access to information in their language, compared to indigenous language-speaking small-scale farmers. We also detected that while cultivating has been traditionally conducted by men, there is a gradual shift toward more active participation by women. The hybrid survey shows that women have increased their participation in the last decades due to decreasing yields of maize in the area, and emigration of men seeking better economic opportunities [66, 67]. This has prompted the involvement of most family members in cultivation activities to increase crop productivity.

## Conclusions, future directions, and limitations

We recommend that local governments and biologists provide practical information to small-scale farmers in the Chiapas Highlands about the importance of the ecosystem services provided by insectivorous bats by feeding on maize crop insect pests. Moreover, there is a need to corroborate whether the higher cost of chemical pesticides is a result of an unnecessary over use by small-scale farmers or simply an increased cost due to transportation of chemicals to remote and isolated communities. In any case, we feel that an integrated pest management program is needed to establish a nature-oriented approach to control crop pests in this region as it has been successful with small-scale farmers in Kenya [68]. It is imperative to shift the perception of some ethnic groups toward the importance of insectivorous bats as active consumers of insect pests damaging their crops. Further, many small-scale farmers are indigenous non-Spanish speakers. The language barrier further reduces their access to information about proper chemical pest management and the importance of insectivorous bats providing ecosystem services. Clearly, it is important to provide the proper information in several autochthonous languages (Table 2).

We also acknowledge some limitations to our study. For example, the ecological niche modeling projected under baseline and future climate change scenarios of species of insectivorous bats in Chiapas did not incorporate other environmental variables, such as landscape connectivity for their dispersal capabilities to reach climatically suitable habitats elsewhere. Further, we assumed that all 67 modeled species of insectivorous bats equally contribute to insect pest control on maize crops and that bats are the only guild of animals involved in insect pest control. We believe that including all species of insectivorous bats ensured that species playing a minor role in consuming insect pests could be relevant to insect pest control if species playing a major role shift their distribution to more climatically favorable habitats. Studies focusing on the relative importance of insectivorous bat species controlling maize insect pests under baseline and future climate change scenarios would provide a risk metric of this environmental service in the region. Second, given the reluctance of some of the communities to participate in surveys/interviews, we applied the snowball technique, in which participants are recommended by other participants [42]. It has been documented that conducting fieldwork and interviewing some ethnic communities in the Chiapas Highlands can be challenging [36]. Nonetheless, we feel that the inclusion of more people in our surveys/interviews has strengthened our results.

## Supporting information

**S1 File. This file contains the structural interview with the following information for each farmer.**
(DOCX)

**S2 File. Excel tables with the geographic information of the ecological niche modeling.** (XLSX)

**S3 File. Excel tables with the answers of the hybrid survey/interview.** (XLSX)

## Acknowledgments

We appreciate the guidance and technical support provided for the economic valuation by Juan Gerardo Juárez Hermosillo and the field support provided by Biologist Isabel Vanessa Flores Sánchez for the hybrid survey.

## Author Contributions

**Conceptualization:** Carolina Ureta, Mercedes Ramírez-Barrón, Felipe Ruán-Soto, Melanie Kolb, Adán L. Martínez-Cruz, Giovanna Gasparello, Víctor Sánchez-Cordero.

**Data curation:** Carolina Ureta, Mercedes Ramírez-Barrón.

**Formal analysis:** Carolina Ureta, Mercedes Ramírez-Barrón, Adán L. Martínez-Cruz.

**Funding acquisition:** Carolina Ureta, Víctor Sánchez-Cordero.

**Investigation:** Carolina Ureta, Mercedes Ramírez-Barrón.

**Methodology:** Carolina Ureta, Mercedes Ramírez-Barrón, Felipe Ruán-Soto, Adán L. Martínez-Cruz.

**Project administration:** Carolina Ureta, Mercedes Ramírez-Barrón.

**Resources:** Carolina Ureta, Felipe Ruán-Soto, Víctor Sánchez-Cordero.

**Software:** Carolina Ureta.

**Supervision:** Carolina Ureta, Felipe Ruán-Soto, Víctor Sánchez-Cordero.

**Validation:** Carolina Ureta.

**Visualization:** Carolina Ureta, Mercedes Ramírez-Barrón.

**Writing – original draft:** Carolina Ureta, Mercedes Ramírez-Barrón, Víctor Sánchez-Cordero.

**Writing – review & editing:** Carolina Ureta, Mercedes Ramírez-Barrón, Felipe Ruán-Soto, Melanie Kolb, Adán L. Martínez-Cruz, Giovanna Gasparello, Víctor Sánchez-Cordero.

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
