## [Decision Letter · Decision Letter 0]

19 Mar 2024

PONE-D-24-03606Impact of Climate Change on Insectivorous Bats and Implications for Small-Scale Farming in southern MexicoPLOS ONE

Dear Dr. Ureta,

Thank you for submitting your manuscript to PLOS ONE. After careful consideration, we feel that it has merit but does not fully meet PLOS ONE’s publication criteria as it currently stands. Therefore, we invite you to submit a revised version of the manuscript that addresses the points raised during the review process.

We look forward to receiving your revised manuscript.

Kind regards,

Raúl Alejandro Alegría-Morán, Ph.D.

Academic Editor

PLOS ONE

Journal Requirements:

"Graduate scholarship from Conahcyt"

5. We note that your Data Availability Statement is currently as follows: The data are contained in the manuscript.

7. Please upload a copy of Supporting Information Figure/Table/etc. "Supplementary Material I" which you refer to in your text on page 10.

**Additional Editor Comments:**

Please follow the comments and suggestions from all the reviewers, particularly reviewer 2 and 4.

Reviewers' comments:

Reviewer's Responses to Questions

**Comments to the Author**

1. Is the manuscript technically sound, and do the data support the conclusions?

Reviewer #1: Partly

Reviewer #2: No

Reviewer #3: Partly

Reviewer #4: No

2. Has the statistical analysis been performed appropriately and rigorously? 

Reviewer #1: No

Reviewer #2: No

Reviewer #3: Yes

Reviewer #4: No

3. Have the authors made all data underlying the findings in their manuscript fully available?

Reviewer #1: Yes

Reviewer #2: No

Reviewer #3: Yes

Reviewer #4: No

4. Is the manuscript presented in an intelligible fashion and written in standard English?

Reviewer #1: Yes

Reviewer #2: No

Reviewer #3: Yes

Reviewer #4: No

5. Review Comments to the Author

Reviewer #1: This article combined quantitative and qualitative methods to conduct an economic evaluation in small-scale field crops in Chiapas considering the potential impacts of climate change on insectivorous bats and the perceptions of small-scale farmers regarding this ecosystem service. The results showed differences in the estimated value of insect pest control by insectivorous bats between official statistics and the author's estimate. Moreover, most interviewed small-scale famers perceive a decline in bat population but are unaware that insectivorous bats consume large proportions of insect pests. Finally, projections of shifts in bat population due to climate change suggest potential declines of insectivorous bats in areas where small-scale farmers cultivate maize for subsistence. Overall, it is a preliminary study for which I have several major comments. Please find them below.

The manuscript would benefit from a clear articulation of the hypothesis that underlies the work. This should be followed up by specific predictions that can be tested using the data collected during the research. The statement of objectives and projections in this Introduction section between line 86 to line 91 does not do an effective job of setting the stage for this research. Please state objectives and projections in a more concise manner. It does not move the work out of a more descriptive context.

In the Methods section, the authors used interview protocol to obtain some information from farmers. The results may not rational because many farmers only know bats a little bit despite the authors employed the snowball sampling and a structured interview. In this case, 53 interviews may not enough.

This study implemented the avoided costs method (ACM) for estimating the economic value of insectivorous bat controlling insect pest on maize crops. And the authors assumed that all species of insectivorous bats contribute equally to pest control and serving as important consumers of insects damaging maize crops. Actually, this assumption may not suitable because each bat species consume different types and numbers of insects. The should be mentioned at least in Discuss section.

In the species distribution models, the authors should consider some variables including land use, human activity and species traits except for climate-related variables.

Reviewer #2: I have carefully examined the manuscript, results, and supplementary material. I am happy to provide my review to the authors in Spanish upon their request or recommendation of the editor.

The authors present a comprehensive three-part study on assigning a monetary value to bat-mediated pest control amid uncertainty on this ecosystem service among local farmers and amid a changing climate. These efforts are vital in addressing negative perceptions of bats and promoting their conservation. A notable strength of this manuscript is the authors' effort to interview farmers, including participants from underrepresented indigenous groups. This approach has the potential to enrich ecological knowledge with insights from traditional knowledge. Additionally, integrating this effort with climate change to predict the future landscape of bat-mediated pest control is rare and valuable in these kinds of qualitative and economic studies.

Despite these merits, the results are often over-interpreted and there are numerous areas where the study and manuscript fall short, leading me to recommend rejection. I urge the authors to thoroughly revisit their dataset to extract deeper insights. I offer the following critiques and suggestions with the hope that they will be constructive in improving future submissions.

Here are my primary concerns:

1. The authors’ interpretation of findings often overlooks methodological limitations, uncertainties, and alternative explanations, with a particular lack of elaboration on the effects of climate change on species richness and participant responses.

2. The background information provided is insufficient. A more detailed introduction (> 3 paragraphs) to the study system, area, local culture, and themes are necessary for context.

3. The literature citations lack expounding, specificity, or relevance to the claims and study area, detracting from the strength of the arguments presented.

4. Interpretations frequently lack justification and relevance to the study system, focusing more on broader disturbances rather than specific data-supported claims.

5. There are inconsistencies and lack of consistent sequence between methods and results, with some results lacking corresponding methods and vice versa. Furthermore, essential data are not provided, such as the percent predicted loss for all bat species. Data that are provided, such as dispersal assumptions, lack clarification, explanation, and justification. I also question the scale and grain of the distribution model and whether the estimates of predicted loss can be explained by model under-prediction for the individual species. All species distribution models have their strengths and weaknesses with respect to over and under-prediction.

6. The authors must make a greater or more nuanced argument for changes in species richness patterns as an indicator for the potential of pest control.

7. The authors primarily conducted in-person surveys rather than qualitative interviews, and the way this was executed introduced confirmation bias. A deeper exploration of the qualitative dataset with conditional grouping and substantiating with participant quotes will be necessary. Given the high sample size of 100 participants (i.e., respondents), it may be worthwhile to design a statistical or sentiment analysis, although sentiment analysis may be complicated given the variety of languages across the study.

8. The findings lack clear conservation implications and recommendations, limiting the practical application of the research or future avenues of study.

9. The manuscript's language could be polished further in English (for English-language journals). I recommend having an English-speaking co-author or colleague copy edit any future iteration of this work before any future journal submissions. It is for this reason, I largely avoid commenting on grammar throughout my review.

Lines 31-33. This claim is not specifically addressed in the body text. The authors should address this counterargument: if climate change will bring more insects, would climate change also attract more insectivorous bats?

Line 45. Not a shift in population but a shift in distribution. Precision needed.

Line 46. The results do not suggest potential decline of bats but rather lower richness in some areas—under certain assumptions. Abundance was not modeled. The authors should elaborate in the body text: to what extent does a loss in richness imply loss of bat-mediated pest control, and therefore an increase in chemical expenditures? For example, bats in the temperate, midwestern US are less diverse than in the neotropics. But US bats forage in areas long-dominated by farmland and are still able to consume vast quantities of insect pests, saving billions of dollars (USD) in pest control.

Lines 49-51. This sentence should be replaced with a conclusion. In what way do these results integrate? What is the broader conclusion of this study? What is the complex interplay? Is it a complex interplay because local farmers are unfamiliar with bat-mediated pest control even though bats provide them at potentially lower cost than purchasing chemicals? This is not complex nor unique. It actually seems straightforward.

Lines 67-68. These are interesting claims but lack detail. For example, why is economic valuation of ecosystem services useful for small-scale agricultural farmers? Have these benefits realized elsewhere? If so, how?

Line 71. Authors must expand on this background. Expand on sociocultural diversity (and maize diversity), key demographics, and provide an overview of the history of maize production in the area (e.g., pre-colonization, colonization, contemporary); also a good place to start referencing historical perspectives on bats, if any, throughout the area or surrounding region. What does the anthropological literature say about this? A quick Google search suggests bats are rooted in Mayan culture. Use this background to highlight the gap in knowledge and synthesize interpretations of the qualitative results.

Lines 74-77. Argue better for the “avoided cost model” and why it was either more superior, more appropriate, or more realistic than the other methods. What do the other methods have in common that make them difficult to obtain similar information from? Experimentation? Long-term monitoring?

Figure 1. The flowchart lacks integration and simplicity. There is no need for GCM nuance. Here, gently introduce the reader to the broader themes. What is the sum of these themes? The “=” symbol in the Economic Valuation portion may mislead readers. Does it mean that pest control chemicals attract bats? Or does it mean that pest control chemicals as useful as insectivory? Perhaps a “vs.” sign can fix this. The PDM column is too complicated. Remove labels and instead use a simplified richness gain/loss visual to show what kind of information you are going to produce for all species. Finally, the caption reads like a method section. No need to use M. cozumelae as an example. In the caption, simply describe what each column represents, from left to right, ending with how the columns integrate. Show a draft of the revised figure to colleagues and non-experts to see if they can interpret it without a caption.

Lines 115-136. There are not enough PDM methods specific to this study. Overall, this section largely regurgitates model fitting from an already published paper, instead of focusing on what was unique about its application in the current study. It is good to tell us what the model is built on and how well it predicts but the reader needs information about how the authors use it. In the results, I was surprised by species-specific gain-loss, predicted species richness, the dispersal assumptions, and would have liked to see explanations and justifications in the methods. This means that the methods are not reproducible. They do not provide enough information for the reader to judge the validity of these results.

I also recommend that the authors use simplified labels for discussing projections under different scenarios throughout the remainder of the text. For example, instead of SSP2-RCP 4.5, call it “moderate development” throughout the remainder of the text, in the figures and the tables. Social scientists and economists will be interested in interpreting the the effects of climate change too and that is the key when producing inter-disciplinary research.

Lines 161-188. This subsection on interview methods lacks justification for the sampling strategy and crucial detail about interviews. Why was snow-ball sampling justified? How difficult is it to access a population of farmers if 100 participants were interviewed? I suspect sampling was opportunistic and not systematically planned.

Instead of referencing a technical protocol for the interview, the authors must provide an overview in the methods of the main text. They must list the questions. They must provide insight into how the responses were coded. The results make it almost seem like this was a survey, not an interview. After inspecting the protocol, it is clear this suspicion was correct. Even some of the open-ended questions are reduced to “yes”, “no”, “I don’t know”. This will need to be reframed as at least a hybrid survey and the authors should take extra care to critically analyze the merit of the questions.

There are fascinating questions in this protocol that I felt were overlooked in the results and synthesis, including education level, religion, “what do bats eat?”, “do you know any stories involving bats and maize?”, and “do you know which kinds of pests are affecting your field?”. Future iterations of this work should clarify how the open-ended responses were coded and integrated into the percentages. For “do you know which kinds of pests are affecting your field?”, you could get a list and determine whether certain bats in the area (or closely related species, or any bat) eat these pests. You could determine if these bats are predicted to leave the area. This would be an example, to me, of integrating the qualitative and quantitative aspects of conservation biology.

The main weakness of the protocol is that it fails to ask the participants “how long have you worked or lived on this specific farmland?” and whether the participants live on these farms or have to travel to them? Without this information, we are unable to interpret question C7: “In recent years, have you seen fewer or more bats in the area?”. Finally, why were participants not asked about climate change and deforestation in the area?

Lines 193-208. These results are intriguing and important for conservation planning. Information for these 67 species should be made publicly available (e.g., Data Dryad) and information for all species should be summarized. For summary, perhaps bin the species into quartiles of predicted suitability loss. Even better, it is important to know the number of species that lose suitable climate space, retain suitable space, or gain suitable climate space. The descriptive narrative of only 11 of the 67 bat species leads me to suspect that most species will not experience substantial loss in suitable climate space under this dispersal assumption, despite being framed as the most concerning. This is cherry picking.

Figure 3. This figure fails to show a discernable trend. I would try something similar to reference 14, where you could set a binary suitability threshold, summarize the loss, gain, and retention. Move the current maps to supplementary. I would also rename the climate scenarios to something more interpretable to a human (see comment for lines 115-136). The biggest issue with this visualization is that it is difficult to see where the colors are changing. It is also difficult to look back and forth from the contemporary baseline to other plots. I challenge the authors to produce a revised visual that the participants of their survey will understand and learn from.

Table 1. I do species distribution modeling and I have no idea what these values mean, what they are based on, nor can I decipher a trend. I doubt most readers will know what this means. Guide the readers through the caption and most importantly, describe this analysis and output in the methods and results.

Line 265. “When asked if participants have observed more or fewer bats in the last year compared to previous [recent] years…”, this question is meaningless. Participants were asked to guess based on data they did not collect, among a population that largely (60%) did not look for bats because they work the farms during the day.

What are previous ["recent"] years? Did the participants elaborate? How long have they tended to these fields? I would give more credence to the 14% that have reported observing them. Their quotes will be helpful here too. I also suspect that the participants were privy to the ecological context of the interview and might have provided an answer they thought the interviewer might have wanted to hear.

Figure 4. This figure title is misleading. Participants were asked what actions they would take if pesticides became too expensive or pests became untenable. The figure title states that these are the actions they would take without bat-mediated pest control. However, this was not the question that was asked.

Line 299-300. This is an interpretation in the results section and should instead be moved to the discussion. It also seems out of place.

Lines 301-305. It should be absolutely clear that this paragraph is describing the monetary evaluation of bat ecosystem services. This is the most important paragraph of the entire results section.

Lines 308-312. Stick to the focus of the study: climate change, bats, and small-scale farms. “Degradation and fragmentation” is too vague and might be referring to loss of vegetation, urbanization, pollution (chemical, atmospheric, sound, light… etc.), or loss of water… etc. Be specific. Farmland contains ample vegetation, water, and clearing for some bats that need more space to maneuver. Is the degradation the authors describe occurring in the outskirts of the farmland?

Line 309. “… combined with a lower presence of insectivorous bats in crops”. This claim is unsubstantiated by the data.

Lines 312-313. This interpretation cannot be trusted because of how the question was asked. See my comment for line 265. This claim also incorrectly references Table 2, a table of ethnic makeup of the participants.

Lines 320-323. This paragraph is difficult to read. Perhaps the authors mean to say that “the ability to respond to changes in the climatic suitability will depend on their ability to disperse, based on available habitat, structural (and functional) connectivity, and flight capabilities”. I also suspect when the authors say “population decrease” they may mean “decrease in richness”? The analysis was based on richness.

Lines 367-368. Reference data and results.

Lines 371-372. How does a lack of understanding about pest services of bats limit strategies on food security, conservation of maize diversity, and bat species richness? What are the strategies? In what ways are these farmers in control of what happens to bat species richness? These would be good to expound upon here and these strategies could be of high conservation value.

Lines 363-366. What could this results mean otherwise?

Reviewer #3: The manuscript titled 'Impact of Climate Change on Insectivorous Bats and Implications for Small-Scale Farming in Southern Mexico' offers valuable insights; however, several revisions are necessary for publication. Notably, the discussion section requires refinement to better integrate the findings, particularly regarding the distribution models of bats. Furthermore, many statements in the manuscript are unsupported by references or there are not enough. Please make sure that whenever a statement is made this is associated with appropriate references.

Most of my comments are in the main manuscript in the attached file.

Reviewer #4: This manuscript presents a compelling study on the ecological interactions between bats and corn pests, highlighting the potential for bat-mediated pest control in agricultural settings. While the premise of the research is intriguing, the execution was not quite there. There are several areas where the manuscript requires stronger scientific rigor, clarity, and impact.

1. Introduction and literature review. The introduction seems to lack a comprehensive review of pertinent literature, missing some crucial studies that could provide a stronger foundation for your research. Ensuring that the introduction succinctly reviews relevant findings and theories will better position your work within the existing scientific discourse. Additionally, the narrative in the introduction appears disjointed, with a structure that leaps between different arguments without a clear, logical progression. Clarifying the research aims and ensuring a coherent flow of ideas will greatly improve the readability and impact of your study.

2. Manuscript Structure. The current structure of the manuscript is trying to cover an overly ambitious range of topics, leading to a treatment of important themes that can feel cursory or inadequately developed. Streamlining the content will ensure that each section is thoroughly and effectively addressed. A clearer, more logical structure, with a well-defined aim presented in the introduction, will help the reader follow your arguments and understand the significance of your research.

3. Methodological Details. The manuscript would benefit from more thorough explanations of the methodology, particularly in relation to how the interactions between bats and corn pests were quantified. A detailed description of the methods used to determine the bats' diet, providing evidence to which of the studies bat species actually consumes the pests, and their impact on pest populations is crucial. The manuscript would benefit from incorporating quantitative methods to assess the extent to which various bat species impact maize pests. This might involve exclusion experiments, bat diet analysis through eDNA, or other ecological and biological data collection methods to quantify the specific contributions of different bat species to pest control. Furthermore, the specific pests affecting corn crops in the study area and the extent of their damage should be comprehensively detailed. This information is pivotal for assessing the validity of your findings. The paper seems to assume uniformly that all insectivorous bats contribute equally to pest control, which is an untested generalization, and overlooks the potential role of other natural enemies of corn pests. This assumption should be critically examined, with a discussion incorporating quantitative methods to substantiate or explore the varied impacts of different bat species on pest populations.

4. Species Distribution Models (SDM). The section on SDMs needs additional information detailing how these models were fitted, the software or packages used, and the rationale behind specific modelling decisions (such as the algorithms and environmental variables used). See specific comments below. It is unclear whether the models were generated specifically for this study or adapted from existing literature. If the former, more detail is needed to describe the models and the assumptions, if the latter, more detail is needed on how the models were adjusted to the different scale of the study. Either way, you should justify the resolution of the models (10km), which seems too broad relative to the study extent.

5. Economic Analysis. The economic analysis presented appears somewhat simplistic. The manuscript should explicitly describe which economic models and approaches were employed in the analysis (e.g., cost-benefit, replacement cost), and the assumptions underlying the economic evaluation. Assess whether the economic analysis adequately considers all relevant costs and benefits, including direct, indirect, and opportunity costs, as well as any potential economic benefits associated with the ecological role of bats in pest control. A detailed explanation of the chosen models, along with their relevance and applicability to the study’s context, is essential for readers to understand the economic implications of the findings.

6. Interview Protocols. The authors need to include a comprehensive description of the interview protocol, outlining the interview process, the exact questions asked, the manner in which the data were recorded and analysed, including any coding strategies, thematic analysis, or statistical methods employed and finally how this data complements the study's overall findings, enhancing the depth and breadth of the research.

Specific Comments:

L48 - revise to "...may force farmers to abandon their crops".

L62-63 - revise to: "...production by preying on large amount of insect pests is widely acknowledged."

L64 - "such as loss..."

Lines 63-68 – Some further detail on the specific aims, objectives and expectations of the research would be useful for the reader here.

Lines 72-82 – Whilst this is useful information and a clear workflow, it is not clear why this figure and text has been placed here in the manuscript. Please incorporate the relevant sections into the Introduction and/or Methods.

Lines 84-90 – Suggest that this section is moved to the end of the Introduction to explain the methodology along with the relevant figure.

Line 92 – please define what a potential distribution model is. Is this the same as an SDM? If so, please outline what these models seek to do and some background on how they work for the reader.

Line 96 – What climatic variables were used in the models? And where was the data sourced from?

Line 98 – Why was a 10-km spacing chosen? And why a minimum of 25 records? Please justify both decisions with text and citations.

Line 100 – What do all these acronyms mean? Please spell out each one in turn and also recognise that not all these are machine-learning algorithms as stated in line 95

Line 101 – Why were these validation metrics chosen? And please explain what they measure and how they work.

Line 103 – Please include citations supporting the biological relevance of the variables used. When was the Pearson test used? After variable selection or on the entire 19 bioclimatic variables? If the latter this is not a suitable method of choosing predictors and I would suggest selecting biologically relevant predictors in the first instance.

Line 106-107 – Are these GCMs from CMIP6? If so, please state this and include the citation. Additionally, please provide citations supporting the use of each GCM.

Line 182: described in Supplementary Material I. This was not provided.

Line 299-300 – Include the scientific names not just the common English and Spanish names of the other invertebrates affecting the crops.

Figure 1: caption contains too much detail for a figure caption.

Figure 2: Needs to have a basemap in the background that shows where Mexico is in relation to other countries. Caption needs further information. Surveyed localities for what? For the interviews?

Figure 3: If you are using a multi panel scheme, in order to be comparable in needs to be the same scale.

Figure 4 & 5 - unsuitable figure format for publication.

6. PLOS authors have the option to publish the peer review history of their article (what does this mean?). If published, this will include your full peer review and any attached files.

Reviewer #1: No

Reviewer #2: No

Reviewer #3: No

Reviewer #4: No

---

## [Author Response · Author response to Decision Letter 0]

25 Jul 2024

Comments from the editor

1. In your Methods section, please provide additional information regarding the permits you obtained for the work. Please ensure you have included the full name of the authority that approved the field site access and, if no permits were required, a brief statement explaining why.

Answer: In the communities interviewed, permission was requested directly and verbally from each of the surveyed farmers in accordance with local customs and traditions. Only in the municipal seat of Teopisca it was necessary to speak with the current Mayor, Rubén de Jesús Valdez Díaz, who also verbally granted permission. We already added this information in our main text and it is highlighted in the methods section. 

"Graduate scholarship from Conahcyt”. Please state what role the funders took in the study. If the funders had no role, please state: ""The funders had no role in study design, data collection and analysis, decision to publish, or preparation of the manuscript." If this statement is not correct you must amend it as needed. 

Answer: Done.

We note that your Data Availability Statement is currently as follows: The data are contained in the manuscript. Please confirm at this time whether or not your submission contains all raw data required to replicate the results of your study. Authors must share the “minimal data set” for their submission. PLOS defines the minimal data set to consist of the data required to replicate all study findings reported in the article, as well as related metadata and methods.

Authors do not need to submit their entire data set if only a portion of the data was used in the reported study. If your submission does not contain these data, please either upload them as Supporting Information files or deposit them to a stable, public repository and provide us with the relevant URLs, DOIs, or accession numbers. For a list of recommended repositories, please see… 

Answer: We already added two more Supplementary materials in order to give access to the information used to carry out our analysis. 

Answer: We will publish all data we used to carry out the analysis. 

8. Please upload a copy of Supporting Information Figure/Table/etc. "Supplementary Material I" which you refer to in your text on page 10.

Answer: We already uploaded that Supplementary material and two more in order to publish all data that were used. We now have three Supplementary material files. 

Reviewer #1: This article combined quantitative and qualitative methods to conduct an economic evaluation in small-scale field crops in Chiapas considering the potential impacts of climate change on insectivorous bats and the perceptions of small-scale farmers regarding this ecosystem service. The results showed differences in the estimated value of insect pest control by insectivorous bats between official statistics and the author's estimate. Moreover, most interviewed small-scale farmers perceive a decline in bat population but are unaware that insectivorous bats consume large proportions of insect pests. Finally, projections of shifts in bat population due to climate change suggest potential declines of insectivorous bats in areas where small-scale farmers cultivate maize for subsistence. Overall, it is a preliminary study for which I have several major comments. Please find them below.

1. The manuscript would benefit from a clear articulation of the hypothesis that underlies the work. This should be followed up by specific predictions that can be tested using the data collected during the research. The statement of objectives and projections in this Introduction section between line 86 to line 91 does not do an effective job of setting the stage for this research. Please state objectives and projections in a more concise manner. It does not move the work out of a more descriptive context.

Answer: Thank you for your feedback. We have incorporated three additional hypotheses into the introduction section, which we believe enhance the clarity of the main text. These hypotheses are now highlighted for easy reference.

2. In the Methods section, the authors used interview protocol to obtain some information from farmers. The results may not be rational because many farmers only know bats a little bit despite the authors employing the snowball sampling and a structured interview. In this case, 53 interviews may not be enough.

Answer: The hybrid survey, as now we defined our methodological approach, was designed to gather information about people's knowledge of bats and their perception regarding their importance and abundance. Although sample size is always critical, and a special analysis considering the target population and the number of variables is always desirable, we consider that as this part of the study is descriptive, it is not necessary to count with a formal analysis of the statistical power (Bolarinwa 2020, Bekele and Ago 2022). Although the objective of this study did not include analyzing data with inferential statistics, even if we did, 53 interviews could be considered sufficient for basic inference. As explained by Memon et al. (2020), over 50 participants could be considered the lower limit for acceptable sample size, even considering the lack of aleatory and independent selection of interview respondents. We included a note in our methods section and highlighted it. The snowball technique was used because in this context, it was not easy to find participants, as indigenous people from the Chiapas Highlands have historically been reluctant to participate in this type of study. This method intended for qualitative work is known to cause biases towards specific types of informants (in this case, farmers with milpas), and was used in this sense in our study with the aim of finding those farmers who were willing to participate in the study. We consider that the point of needing a higher sample size to counteract the introduced biases (Memon et al. 2020) is not valid in our case. We already added information in the methods and discussion sections. This new information has been highlighted. 

3. This study implemented the avoided costs method (ACM) for estimating the economic value of insectivorous bat controlling insect pest on maize crops. And the authors assumed that all species of insectivorous bats contribute equally to pest control and serving as important consumers of insects damaging maize crops. Actually, this assumption may not suitable because each bat species consume different types and numbers of insects. The should be mentioned at least in Discuss section.

Answer: Thank you for your comment and we completely agree with you. We already added our assumption in the methods and discussion sections.

4. In the species distribution models, the authors should consider some variables including land use, human activity and species traits except for climate-related variables.

Answer: Although it would have been ideal to integrate as much information as possible to the ecological niche modeling, given that the main focus of the present study is climate change, the variables used would have needed to have a projected layer as a function of climate. Otherwise, we would have needed to assume that they would remain identical in the forthcoming decades. This assumption increases the uncertainty of the modeling. That is why several studies working with ecological niche modeling and climate change do not use other environmental or social variables in their analysis (Ureta et al. 2018, Ureta et al. 2020, Velasco et al. 2021, Ureta et al. 2022a). However; in the main text we clarified that we are working with the climatic niche only. Also, the maps used have been modeled in another study and we only used the species of interest that are the ones existing in Chiapas. 

Reviewer #2: I have carefully examined the manuscript, results, and supplementary material. I am happy to provide my review to the authors in Spanish upon their request or recommendation of the editor.

The authors present a comprehensive three-part study on assigning a monetary value to bat-mediated pest control amid uncertainty on this ecosystem service among local farmers and amid a changing climate. These efforts are vital in addressing negative perceptions of bats and promoting their conservation. A notable strength of this manuscript is the authors' effort to interview farmers, including participants from underrepresented indigenous groups. This approach has the potential to enrich ecological knowledge with insights from traditional knowledge. Additionally, integrating this effort with climate change to predict the future landscape of bat-mediated pest control is rare and valuable in these kinds of qualitative and economic studies. Despite these merits, the results are often over-interpreted and there are numerous areas where the study and manuscript fall short, leading me to recommend rejection. I urge the authors to thoroughly revisit their dataset to extract deeper insights. I offer the following critiques and suggestions with the hope that they will be constructive in improving future submissions.

Here are my primary concerns:

Answer: Thank you for your comments. We are sure our ms will be a better version when taken all of them into consideration. 

1. The authors’ interpretation of findings often overlooks methodological limitations, uncertainties, and alternative explanations, with a particular lack of elaboration on the effects of climate change on species richness and participant responses.

Answer: We already tried to have a more detailed explanation about the methodological limitations and their consequences on our results. We rephrased the entire Discussion section. 

2. The background information provided is insufficient. A more detailed introduction (> 3 paragraphs) to the study system, area, local culture, and themes are necessary for context.

Answer: We already expanded the introduction and deepened into the study system, area and local culture. Please find the new information highlighted in the introduction section. 

3. The literature citations lack expounding, specificity, or relevance to the claims and study area, detracting from the strength of the arguments presented.

Answer: We already revised all the literature cited. 

4. Interpretations frequently lack justification and relevance to the study system, focusing more on broader disturbances rather than specific data-supported claims.

Answer: We rephrased about 70% of our main text in order to justify our entire discussion section and we based our ms in our results only.

5. There are inconsistencies and lack of consistent sequence between methods and results, with some results lacking corresponding methods and vice versa. Furthermore, essential data are not provided, such as the percent predicted loss for all bat species. Data that are provided, such as dispersal assumptions, lack clarification, explanation, and justification. I also question the scale and grain of the distribution model and whether the estimates of predicted loss can be explained by model under-prediction for the individual species. All species distribution models have their strengths and weaknesses with respect to over and under-prediction.

Answer: Thank you for your comment. We already added several detailed information in our methods and results sections. We believe our entire ms is clearer now. Also, we integrated two more Supplementary material files and changed our figures. 

6. The authors must make a greater or more nuanced argument for changes in species richness patterns as an indicator for the potential of pest control.

Answer: We decided we will not integrate species richness anymore as it gets confusing across the reading. We wanted to understand the risk that climate change might represent for the insectivorous bats in Chiapas in particular, but do not really need to evaluate the richness. We rephrased several sections and stated three different hypotheses (as suggested by a reviewer) that were our guide in the entire ms. 

7. The authors primarily conducted in-person surveys rather than qualitative interviews, and the way this was executed introduced confirmation bias. A deeper exploration of the qualitative dataset with conditional grouping and substantiating with participant quotes will be necessary. Given the high sample size of 100 participants (i.e., respondents), it may be worthwhile to design a statistical or sentiment analysis, although sentiment analysis may be complicated given the variety of languages across the study.

Answer: Although our methodological design was based as a structured interview (sensu Bernard 2017), we decided to call it hybrid survey as suggested by this revision given that there were two different types of questions. However; we did our questions face to face and respondents were free to answer whatever they wanted in some of the questions. We believe there is no confirmation bias. Also, several questions were freely responded, so that this information could later be captured and categorized by the authors according to the study's objectives. On the other hand, sentiment analysis was not considered useful according to the hypotheses of this study, as we sought to understand people's perception of bats and their importance as pest controllers.

8. The findings lack clear conservation implications and recommendations, limiting the practical application of the research or future avenues of study.

Answer: We now clarified the implications of the study and give recommendations in our conclusions section. 

9. The manuscript's language could be polished further in English (for English-language journals). I recommend having an English-speaking co-author or colleague copy edit any future iteration of this work before any future journal submissions. It is for this reason, I largely avoid commenting on grammar throughout my review.

Answer: Thank you for your recommendation, we send our ms to a native speaker editor before submitting our revised version. 

10. Figure 1. The flowchart lacks integration and simplicity. There is no need for GCM nuance. Here, gently introduce the reader to the broader themes. What is the sum of these themes? The “=” symbol in the Economic Valuation portion may mislead readers. Does it mean that pest control chemicals at

---

## [Decision Letter · Decision Letter 1]

23 Aug 2024

PONE-D-24-03606R1Impact of Climate Change on the Distribution of Insectivorous Bats: Implications for Small-Scale Farming in Southern MexicoPLOS ONE

Dear Dr. Ureta,

Thank you for submitting your manuscript to PLOS ONE. After careful consideration, we feel that it has merit but does not fully meet PLOS ONE’s publication criteria as it currently stands. Therefore, we invite you to submit a revised version of the manuscript that addresses the points raised during the review process.

Please take particular care for the suggestions and comments from Reviewers #2 and #3. 

We look forward to receiving your revised manuscript.

Kind regards,

Raúl Alejandro Alegría-Morán, Ph.D.

Academic Editor

PLOS ONE

Journal Requirements:

Reviewers' comments:

Reviewer's Responses to Questions

**Comments to the Author**

1. If the authors have adequately addressed your comments raised in a previous round of review and you feel that this manuscript is now acceptable for publication, you may indicate that here to bypass the “Comments to the Author” section, enter your conflict of interest statement in the “Confidential to Editor” section, and submit your "Accept" recommendation.

Reviewer #1: All comments have been addressed

Reviewer #2: (No Response)

Reviewer #3: All comments have been addressed

2. Is the manuscript technically sound, and do the data support the conclusions?

Reviewer #1: Yes

Reviewer #2: Yes

Reviewer #3: Yes

3. Has the statistical analysis been performed appropriately and rigorously? 

Reviewer #1: Yes

Reviewer #2: Yes

Reviewer #3: Yes

4. Have the authors made all data underlying the findings in their manuscript fully available?

Reviewer #1: Yes

Reviewer #2: Yes

Reviewer #3: Yes

5. Is the manuscript presented in an intelligible fashion and written in standard English?

Reviewer #1: Yes

Reviewer #2: Yes

Reviewer #3: Yes

6. Review Comments to the Author

Reviewer #1: (No Response)

Reviewer #2: I reviewed the first iteration of this manuscript and applaud the authors for their great work in addressing the reviewers’ critiques and suggestions. I previously noted major concerns about the format of the questions and responses in the participant study – and in particular the conclusions that the authors drew from them. Now, the interpretations are much more careful, intriguing, and I feel they can galvanize conservation efforts in the area. I really liked that the authors’ argued for the validity of the farmers’ perceived decline in bats and that if these long-standing farmers are recognizing fewer bats, then we need to start monitoring and collecting data in the area immediately. I was also enlightened by the added cultural background and local context. Dropping the richness estimates improved the relevance of the SDM. The additional supplementary data was also very helpful and allowed me to better understand the study’s findings. As a result, I feel the spirit and substance of the manuscript is now integrated as per the authors’ original intent.

I still feel there are still some areas that can improve the interpretability of the manuscript.

Table 1 caption: While much improved from the original manuscript, the caption must be able to stand alone from the main text. First, in the caption, re-explain what the GCMs indicate (as per lines 135-141). Clarify that these values represent percentages of pixel loss/gain relative to contemporary projections. Re-clarify the spatial grain: what each pixel represents. Making these changes will help your readers immensely.

Table 1 format: I don’t believe we need both the mean and median. Pick one measure of centrality that most honestly fits with the distribution of the data. The other measure will still be in the supplementary data. If the data are not a normal distribution, use the median (and vice versa). The current table’s `Range change` column is redundant, containing values that are already in the same row so it can be removed. If the authors’ still want to direct the readers to minimum and maximum values, the values in each row can be bolded or italicized (and specified in the caption). Finally, I liked the format of the table in SupplementaryMaterial_II in the `Total`tab better. It is much more transparent with pixel information so I recommend using the tabular format of one of those tables. The authors must indicate that the `%Loss` column reflects the non-dispersal scenario and `Range change%`column reflects the limited-full dispersal column. The reason suppmat format works is because GCMs are the rows and the descriptive statistics are the columns.

Alternatively, consider pivoting the years for each GCM into descriptive statistic columns, if possible. That way the reader reads time from left to right. Still pick either the median or mean, depending on the distribution of the data.

Lines 333 – 335: I recommend that the authors also include dietary monitoring as a conservation need, given the dearth of dietary information for species in the area and relevance to the study’s topic. DNA metabarcoding has made dietary screening more accessible than ever before. If bats are ever mist-netted, it is no issue to collect guano and send for DNA screening, so long as funding proposals include this as a goal. There are papers that have even used the DNA metabarcoding to demonstrate pest consumption as an ecosystem service in areas with small-scale agriculture and I don’t think it will be difficult for the authors to find a few either in their surrounding regions or others, like African nations, for example. Metabarcoding not only allows for pest detection but allows acquisition of dietary diversity so that changes in consumption patterns can be monitored. Perhaps, dietary data could also clarify other perceptions of the farmers (e.g., are the bats also eating their corn crop?).

Minor comments:

Line 46: “Surveyed farmers perceived a decline in bat populations” – percentage needed.

Lines 68-69: This sentence is out of place. I would delete it or move and integrate into the final paragraph of the introduction.

Lines 75 – 77: Consider starting a new paragraph with this sentence because it functions as a topic sentence for the remaining sentences on the local society and culture.

Lines 80 – 87: These sentences should be moved and integrated into the final paragraph of the introduction.

Line 102: delete “there”

Line 105: replace comma with semicolon.

Line 318: $16’899,132 – replace apostrophe with comma

Line 325: Delete “do”

Lines 327 – 328: Sentence is a bit wordy and difficult for me to follow. Consider the following line correction: “Insectivorous bat distributions under both climate scenarios will likely have negative impacts …” Or determine another revision for readability.

Lines 349 - 351: I do not understand this sentence. Revise for clarity.

Reviewer #3: (No Response)

7. PLOS authors have the option to publish the peer review history of their article (what does this mean?). If published, this will include your full peer review and any attached files.

Reviewer #1: No

Reviewer #2: No

Reviewer #3: No

---

## [Author Response · Author response to Decision Letter 1]

27 Aug 2024

1. Reviewer #2: I reviewed the first iteration of this manuscript and applaud the authors for their great work in addressing the reviewers’ critiques and suggestions. I previously noted major concerns about the format of the questions and responses in the participant study – and in particular the conclusions that the authors drew from them. Now, the interpretations are much more careful, intriguing, and I feel they can galvanize conservation efforts in the area. I really liked that the authors’ argued for the validity of the farmers’ perceived decline in bats and that if these long-standing farmers are recognizing fewer bats, then we need to start monitoring and collecting data in the area immediately. I was also enlightened by the added cultural background and local context. Dropping the richness estimates improved the relevance of the SDM. The additional supplementary data was also very helpful and allowed me to better understand the study’s findings. As a result, I feel the spirit and substance of the manuscript is now integrated as per the authors’ original intent.

I still feel there are still some areas that can improve the interpretability of the manuscript.

Answer: Thank you for your time and feedback that helped improve our manuscript. 

2. Table 1 caption: While much improved from the original manuscript, the caption must be able to stand alone from the main text. First, in the caption, re-explain what the GCMs indicate (as per lines 135-141). Clarify that these values represent percentages of pixel loss/gain relative to contemporary projections. Re-clarify the spatial grain: what each pixel represents. Making these changes will help your readers immensely.

Answer: We tried to clarify the Table caption to make it stand alone. We highlighted the Table caption in order to facilitate the view of corrections. 

Table 1 format: I don’t believe we need both the mean and median. Pick one measure of centrality that most honestly fits with the distribution of the data. The other measure will still be in the supplementary data. If the data are not a normal distribution, use the median (and vice versa). The current table’s `Range change` column is redundant, containing values that are already in the same row so it can be removed. If the authors’ still want to direct the readers to minimum and maximum values, the values in each row can be bolded or italicized (and specified in the caption). 

Answer: Done. 

Finally, I liked the format of the table in SupplementaryMaterial_II in the `Total`tab better. It is much more transparent with pixel information so I recommend using the tabular format of one of those tables. The authors must indicate that the `%Loss` column reflects the non-dispersal scenario and `Range change%`column reflects the limited-full dispersal column. The reason suppmat format works is because GCMs are the rows and the descriptive statistics are the columns.

Alternatively, consider pivoting the years for each GCM into descriptive statistic columns, if possible. That way the reader reads time from left to right. Still pick either the median or mean, depending on the distribution of the data.

Answer: We changed the table as suggested. We hope it is clearer now. 

Lines 333 – 335: I recommend that the authors also include dietary monitoring as a conservation need, given the dearth of dietary information for species in the area and relevance to the study’s topic. DNA metabarcoding has made dietary screening more accessible than ever before. If bats are ever mist-netted, it is no issue to collect guano and send for DNA screening, so long as funding proposals include this as a goal. There are papers that have even used the DNA metabarcoding to demonstrate pest consumption as an ecosystem service in areas with small-scale agriculture and I don’t think it will be difficult for the authors to find a few either in their surrounding regions or others, like African nations, for example. Metabarcoding not only allows for pest detection but allows acquisition of dietary diversity so that changes in consumption patterns can be monitored. Perhaps, dietary data could also clarify other perceptions of the farmers (e.g., are the bats also eating their corn crop?).

Answer: Thank you for your comment. It is a great goal and would be amazing to achieve it. We incorporated this information in our text and added the following reference:

Gonçalves, A., Nóbrega, E. K., Rebelo, H., Mata, V. A., & Rocha, R. (2024). A metabarcoding assessment of the diet of the insectivorous bats of Madeira Island, Macaronesia. Journal of Mammalogy, 105(3), 524-533.

It would be wonderful to think in a future collaboration if you might be interested. 

Minor comments:

Line 46: “Surveyed farmers perceived a decline in bat populations” – percentage needed.

Answer: Done

Lines 68-69: This sentence is out of place. I would delete it or move and integrate into the final paragraph of the introduction.

Answer: We deleted it. 

Lines 75 – 77: Consider starting a new paragraph with this sentence because it functions as a topic sentence for the remaining sentences on the local society and culture.

Answer: Done. 

Lines 80 – 87: These sentences should be moved and integrated into the final paragraph of the introduction.

Answer: We rephrased following your advice. 

Line 102: delete “there”

Answer: Done. 

Line 105: replace comma with semicolon.

Answer: Done. 

Line 318: $16’899,132 – replace apostrophe with comma

Answer: Done. 

Line 325: Delete “do”

Answer: Done. 

Lines 327 – 328: Sentence is a bit wordy and difficult for me to follow. Consider the following line correction: “Insectivorous bat distributions under both climate scenarios will likely have negative impacts …” Or determine another revision for readability.

Answer: Done. 

Lines 349 - 351: I do not understand this sentence. Revise for clarity.

Answer: Done.

---

## [Decision Letter · Decision Letter 2]

4 Sep 2024

Impact of Climate Change on the Distribution of Insectivorous Bats: Implications for Small-Scale Farming in Southern Mexico

PONE-D-24-03606R2

Dear Dr. Ureta,

We’re pleased to inform you that your manuscript has been judged scientifically suitable for publication and will be formally accepted for publication once it meets all outstanding technical requirements.

Kind regards,

Raúl Alejandro Alegría-Morán, Ph.D.

Academic Editor

PLOS ONE

Additional Editor Comments (optional):

Reviewers' comments:

Reviewer's Responses to Questions

**Comments to the Author**

1. If the authors have adequately addressed your comments raised in a previous round of review and you feel that this manuscript is now acceptable for publication, you may indicate that here to bypass the “Comments to the Author” section, enter your conflict of interest statement in the “Confidential to Editor” section, and submit your "Accept" recommendation.

Reviewer #2: All comments have been addressed

2. Is the manuscript technically sound, and do the data support the conclusions?

Reviewer #2: Yes

3. Has the statistical analysis been performed appropriately and rigorously? 

Reviewer #2: Yes

4. Have the authors made all data underlying the findings in their manuscript fully available?

Reviewer #2: Yes

5. Is the manuscript presented in an intelligible fashion and written in standard English?

Reviewer #2: Yes

6. Review Comments to the Author

Reviewer #2: (No Response)

7. PLOS authors have the option to publish the peer review history of their article (what does this mean?). If published, this will include your full peer review and any attached files.

Reviewer #2: No

---

## [Editor Report · Acceptance letter]

17 Oct 2024

PONE-D-24-03606R2 

PLOS ONE

Dear Dr. Ureta, 

I'm pleased to inform you that your manuscript has been deemed suitable for publication in PLOS ONE. Congratulations! Your manuscript is now being handed over to our production team.

Kind regards, 

on behalf of

Dr. Raúl Alejandro Alegría-Morán 

Academic Editor

PLOS ONE